

# Does how individuals handle social situations exacerbate the relationship between physique anxiety and food addiction? The role of emotional expressive suppression and social avoidance and distress

Yan Li, Yuxia Fan, Jing Lin and Shaobo Shi

Qingdao Mental Health Center, Qingdao, Shandong, China

## ABSTRACT

**Background**. Research on food addiction has increased significantly in recent years. It has been demonstrated that food addiction can lead to impairments in physiological, psychological, and social functioning in individuals. However, there is a lack of studies investigating the influence of how individuals handle social situations on food addiction and the specific mechanisms involved.

**Method**. A cross-sectional survey was conducted with 1,151 university students, with a mean age of 21.44 ($SD = 4.77$) years. The sample comprised 74.46% female and 25.54% male students. Participants completed the Chinese version of the modified Yale Food Addiction Scale 2.0, the Social Physique Anxiety Scale, the Expressive Suppression Scale, and the Social Avoidance and Distress Scale. Statistical analyses were performed using SPSS 26.0 and the Process (Version 3.4) plug-in.

**Result**. The results of the study supported our hypothesis that the association between social physique anxiety and food addiction symptoms could be partially explained by expressive suppression and social avoidance and distress. This association remained significant even after adjusting for covariates such as gender, number of cigarettes smoked per day, bedtime, education, and BMI. Specifically, more severe social physique anxiety was found to be associated with frequent use of expressive suppression and social avoidance and distress, which in turn was associated with more severe food addiction symptoms.

**Conclusion**. This study explored the role of expression suppression and social avoidance and distress in the relationship between social physique anxiety and food addiction symptoms. The findings provide a theoretical basis for developing interventions for food addiction in college students. These interventions could include helping students develop a healthy perception of body image, encouraging emotional expression, and promoting active social participation to reduce food addiction symptoms.

Corresponding author
Shaobo Shi, ssb7129@163.com

## INTRODUCTION

The concept of food addiction was initially introduced in 1956, described as "a common pattern of symptoms, akin to those observed in other addictive processes" (*Randolph, 1956*). According to the diagnostic criteria for substance dependence, food addiction encompasses the inability to exert rational control over the consumption quantity and duration of food intake, coupled with the emergence of behaviors such as tolerance, withdrawal, and intense cravings, particularly for highly processed, calorie-dense foods (*Gearhardt, Corbin & Brownell, 2009*; *Ifland et al., 2009*). Food addiction, often categorized as an eating disorder, shares genetic markers with other similar disorders, including those that code for the dopamine D2 receptor and the opioid receptor gene (OPRM1 gene) (*Davis et al., 2011*) with studies indicating symptomatic convergence between food addiction and other eating disorders through questionnaire-based assessments (*Ratković et al., 2023*; *Walenda et al., 2021*). Food addiction can precipitate a plethora of adverse health outcomes. These include psychological or psychiatric disturbances (*e.g.*, despondency, weight gain culminating in diminished self-esteem, severe depression, and binge eating disorder), physical ailments (*e.g.*, obesity or overweight status, metabolic dysregulation due to selective eating habits, diabetes, or cardiovascular diseases), and social issues (*e.g.*, apprehension over stigmatization attributable to excessive weight or behaviors resembling addiction) (*Borisenkov, Tserne & Bakutova, 2018*; *De Almeida, Kamath & Cabandugama, 2022*; *Ivezaj, Wiedemann & Grilo, 2017*; *Pavanello et al., 2022*; *Rose, Nadler & Mackey, 2018*).

During the college years, an individual's thoughts, behaviors, and emotions gradually develop to maturity—that is, personality development and maturation (*Lipsky et al., 2024*). In this particular phase of life, a colorful dietary environment is presented to college students, and at the same time, this phase is important for the formation of individual dietary preferences (*Bogár, Pászthy & Túry, 2024*). The heightened milieu of living arrangements, academic demands, and responsibilities predisposes college students more than other population to a higher likelihood of developing eating disorders (*Kyrkou et al., 2018*; *Stok et al., 2018*), including food addiction (*Pursey et al., 2014*). Studies have revealed a notable prevalence of food addiction among Chinese college students, reaching up to 13.24%, which is in the middle of the prevalence of food addiction in non-clinical samples from global surveys (8.2%–22.2%) (*Penzenstadler et al., 2019*). However, which factors and underlying mechanisms contribute to food addiction remains inadequately addressed within Chinese scholarly discourse (*Luo et al., 2022*). While existing research underscores the nexus between food addiction and obesity, certain inquiries have been raised regarding this correlation, necessitating further examination into their interplay (*Hebebrand & Gearhardt, 2021*). For instance, the deleterious effects of food addiction extend beyond obesity, encompassing psychological distress and compromised social functioning (*Burrows et al., 2018*). Therefore, food addiction screening college students with both obesity problems and those without is necessary to explore the food addiction of Chinese college students.

## Social physique anxiety and food addiction

Anxiety, the most diagnosed mental disorder among Chinese individuals, has a lifetime prevalence as high as 7.57% (*Huang et al., 2019*), which is higher than the global population lifetime prevalence of anxiety disorders (7.3%) (*Baxter et al., 2013*). There is well-established research confirming a correlation between anxiety and food addiction (*da Silva Júnior et al., 2022*; *Fekih-Romdhane et al., 2022*; *Hussenoeder et al., 2022*; *Nolan & Jenkins, 2019*). Social physique anxiety, a specific category of anxiety, is characterized by apprehension surrounding one's physical appearance in social contexts due to uncertainty about making a favorable impression (*Xu, 2003*). The prevalence of social physique anxiety among Chinese college students may be partially attributed to the cultural idealization of slimness as beauty and the pervasive influence of this ideal on social networks (*Fu et al., 2022*; *Hurst et al., 2000*). Within the realm of social media, students often engage in subconscious comparison of their bodies with those of others. This can lead to many negative behaviors, such as unhealthy eating habits and addictive behaviors, as means to achieve a coveted physique, particularly when they perceive themselves as less fit in comparison to their peers others (*Thøgersen-Ntoumani et al., 2017*). To mitigate social physique anxiety, individuals frequently reduce their food intake and increase their physical activity to attain a desired body shape, potentially leading to eating disorders or exercise disorders (*Alcaraz-Ibáñez, Paterna & Griffiths, 2023*; *Cook et al., 2015*). The potential link between social physique anxiety and eating disorders is conceivable, that is, striving to evade negative social scrutiny about one's body may prompt specific problematic eating behaviors (*Kowalski et al., 2006*; *Lanfranchi et al., 2015*). Nonetheless, the exact nature of the relationship between social physique anxiety on individual dietary choices is yet to be clarified (*Alcaraz-Ibáñez, Paterna & Griffiths, 2023*) and warrants further examination.

Nevertheless, some research has established a indirect correlation between social physique anxiety and food addiction. For instance, a study encompassing 555 Spanish university students identified a positive correlation between social physique anxiety and disordered eating patterns, such as bulimia and persistent food-related thoughts (*Alcaraz-Ibáñez et al., 2020*). Similarly, research involving 766 French adolescents elucidated how social physique anxiety contributes to detrimental eating attitudes and behaviors in adolescents (*Lanfranchi et al., 2015*). A recent meta-analytical review uncovered a robust overall association between social physique anxiety and eating disorders, quantified by a correlation coefficient of 0.5 (*Alcaraz-Ibáñez, Paterna & Griffiths, 2023*). Based on the evidence presented, it is plausible to deduce that social physique anxiety may exert a direct positive influence on the propensity for food addiction.

## Role of expressive suppression

Emotion regulation strategies constitute the mechanisms through which individuals exert control over the occurrence, intensity, and expression of their emotions (*Thompson, 1994*). Based on Gross's theory of emotion regulation, expressive suppression emerges as a pivotal emotion regulation strategy, typically manifesting in the latter stages of the emotional process. It attenuates the subjective experience of emotion, predominantly by curbing the impending or active emotional expressions (*Gross, 2002*). Expressive suppression is linked

to a spectrum of psychological distress, being associated with mood disturbances such as anxiety and depression. Individuals plagued by mood disorders often exhibit heightened levels of expressive suppression (*Lincoln, Schulze & Renneberg, 2022*; *Liu et al., 2022*). Concurrently, social physique anxiety is rooted in an individual's anguish related to their body image within social contexts, deriving from body image discontent (*Jin & Fung, 2021*; *Swami, Robinson & Furnham, 2021*; *Zartaloudi et al., 2023*), Previous investigations have revealed that such dissatisfaction propels individuals toward expressive suppression (*Wang, Liu & Lei, 2023*), leading to the hypothesis that social physique anxiety and expressive suppression are interconnected. Extreme emotional states may prompt problematic eating behaviors, which are intricately associated with disorders of emotional regulation (*Brunault & Ballon, 2021*). Nonadaptive emotion regulation is now recognized as a core component of treatment for addictive disorders (*Prefit, Cândea & Szentagotai-Tătar, 2019*). Those who engage in nonadaptive emotion regulation are more inclined to impulsive reactions under emotionally charged and disagreeable circumstances (*Wolz et al., 2016*), Under such pressure, they may resort to food as a means to mitigate their emotional turmoil, particularly those with a high propensity toward food addiction (*Gearhardt, Corbin & Brownell, 2016*; *Ribeiro et al., 2023*). Hence, food addiction is regarded as a maladaptive emotion regulation tactic. It is postulated that expressive suppression could act as a mediating factor in the relationship between social physique anxiety and food addiction.

## Role of social avoidance and distress

Social avoidance and distress encompass both the behavioral proclivity to eschew social interaction and the accompanying emotional turmoil experienced in actual social settings (*Watson & Friend, 1969*). College students grappling with social physique anxiety often report concomitant dissatisfaction with their physical appearance (*Jin & Fung, 2021*; *Swami, Robinson & Furnham, 2021*; *Zartaloudi et al., 2023*). Research indicates that the more acute the self-critical view of one's physique, the greater the likelihood of such students experiencing social avoidance and distress in social environments (*Miers et al., 2014*). A principal aspect of social physique anxiety involves trepidation over adverse judgments concerning one's physique and the undue scrutiny from others (*Zartaloudi et al., 2023*), and cognitive-behavioral model suggests that the confluence of fear of negative evaluation and heightened self-awareness is likely to engender social trepidations (*Rapee & Heimberg, 1997*). Fears of adverse appraisals may lead individuals to experience heightened vigilance, withdraw from social endeavors, and even cultivate diminished self-esteem in social contexts (*Perczel-Forintos & Kresznerits, 2017*; *Willemse et al., 2023*). Consequently, we posit that social physique anxiety has a direct and positive impact on social avoidance and distress among college students.

## The role of expressive suppression and social avoidance and distress

Furthermore, social avoidance and distress is implicated in augmenting negative emotions and undermining social support, as a result of reduced social engagements (*Chen et al., 2023*; *Li et al., 2023*). As per the reinforcement theory of negative emotions, evasion of adverse emotions is foundational to the emergence of addictive behaviors (*Baker et al.,*

*2004*). To regulate the negative emotions and diminished emotional support associated with social avoidance and distress, students may turn to eating—a behavior known to furnish gratification—as a means of emotional and stress alleviation. Thus, suggesting a linkage between social avoidance and distress and food addiction. The emotion regulation process model indicates that the aim of emotional regulation is to reconcile conflicting demands by selecting responses that are deemed socially acceptable in varied circumstances, and difficulties in any aspect of emotional regulation may precipitate social adjustment challenges (*Gross, 2002*). Hence, college students dealing with social physique anxiety who engage in maladaptive emotion regulation strategies may be predisposed to develop social avoidance and distress (*Campos et al., 1994*). Therefore, we hypothesized that social avoidance and distress mediates the association between social physique anxiety and food addiction, with both expressive suppression and social avoidance and distress acting in serial mediation between social physique anxiety and food addiction.

### The present study

In summary, this study aims to examine the potential roles of expressive suppression and social avoidance and distress in social physique anxiety and food addiction. As depicted in Fig. 1, we propose a conceptual model involving expressive suppression and social avoidance and distress (*i.e.,* social physique anxiety → expressive suppression → social avoidance and distress → food addiction). Specifically, we hypothesize that heightened levels of social physique anxiety correlate with increased utilization of expressive suppression as a means of emotion regulation, potentially leading to elevated food addiction (H1). Additionally, we posit that more pronounced social physique anxiety is associated with heightened social avoidance and distress, which in turn is linked to increased food addiction (H2). Furthermore, we propose that expressive suppression and social avoidance and distress function as sequential process factors contributing to elevated social physique anxiety and food addiction (H3).

## METHOD

### Participants

The investigation was conducted using a convenience sampling methodology across three universities in Qingdao, within the Shandong Province of China. From an initial pool of 1,300 questionnaires disseminated, a total of 1,151 were recognized as valid for analysis, after the exclusion of 149 for reasons including random and patterned responses, culminating in an efficacy rate of 88.53%. Eligibility for participation was restricted to university students possessing the cognitive ability to understand and respond to the survey instrument, contingent upon their informed consent. Exclusions applied to individuals who had experienced traumatic events within the last year or had been on a sabbatical. Prior to the commencement, requisite approvals were obtained from the concerned academic authorities. The purpose of the study was transparently communicated to the participants, namely to explore the relationship between social physique anxiety and food addiction within the student demographic. Ethical assurances were underscored: this study is anonymous and the prerogative to withdraw remained with the participants

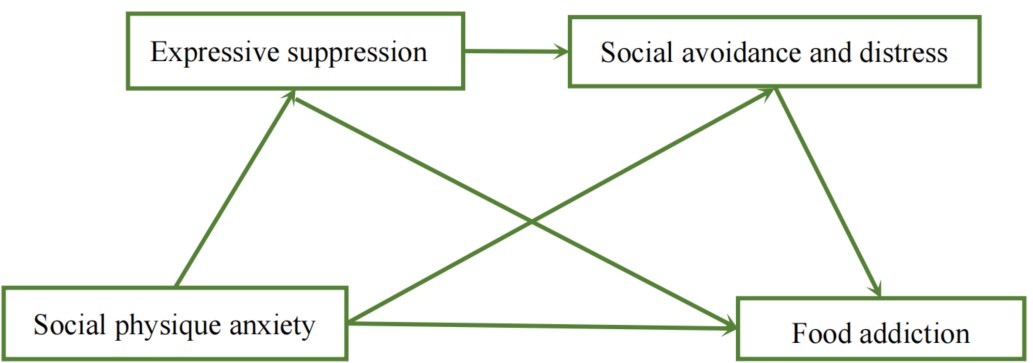

**Figure 1  The conceptual model for food addiction, social physique anxiety, expressive suppression, social avoidance and distress.**

at any point; the collected data were exclusively for scholarly analysis and confined to the scrutiny of the research cohort. The integrity and sincerity of the responses were of paramount importance, with participants exhorted to answer truthfully and in accordance with their actual perceptions and conduct. The survey instrument was congruent with those employed in previous inquiries, deliberately avoiding any ethically dubious items. The process entailed participants reviewing an electronic informed consent document before engaging with the online survey instrument (Questionnaire Star) on a voluntary basis. The ethical approbation for this study was accorded by the Ethics Committee of the Qingdao Mental Health Center (QDJWZXWZLL2024002), in strict accordance with the principles established by the Declaration of Helsinki.

## Measurements
### General demographics
Gender, age, smoking, bedtime, education, and BMI were included as covariates based on previous studies of food addiction (*Arslan et al., 2024*; *Bartschi & Greenwood, 2023*; *Hoover et al., 2023*; *Niroumand Sarvandani et al., 2024*). So we designed the participant general demographics which included sex (0 = male, 1 = female), age, BMI (weight (kg)/height (m)$^2$, (1 = < 18.5, 2 = 18.5–23.9, 3 = 24–27.9, 4 = > 28)), home location (0 = city, 1 = country), bedtime (1 = before 22: 00, 2 = 22: 00 –24: 00, 3 = after 24: 00), educational (1 = 3–year college student, 2 = undergraduate college student, 3 = postgraduate college student) and cigarettes per day (0 = no smoking, 1 = 1 day < 10 cigarettes, 2 = 1 day 10–20 cigarettes, 3 = 1 day > 20 cigarettes).

### The Chinese modified Yale food addiction scale 2.0
The Chinese modified Yale Food Addiction Scale 2.0 (C-mYFAS2.0) is a 13-item self-report scale designed to assess food addiction status over the past year (*Zhang et al., 2021*). The original authors have granted a license for its use within the all Chinese population (*Schulte & Gearhardt, 2017*). Two items (*e.g.*, "My eating behavior caused me much distress") evaluate diet-related clinical impairment and distress, while eleven items (*e.g.*, "I ate to the point where I felt physically ill") gauge symptoms of food addiction over the previous

twelve months. Each item is scored dichotomously according to the threshold established by the C-mYFAS2.0 validation study (0 = criterion not met, 1 = criterion met). This criterion is supported if any item aligns with the diagnostic criteria or if the clinical severity meets the specified threshold.

The C-mYFAS2.0 can be scored in two ways: the Symptom-count version, which sums all but the two items measuring diet-related clinical impairment or distress, yielding total scores ranging from 0 to 11, with higher scores indicating a greater propensity for food addiction. The second method is the hierarchical diagnostic approach, which sums the scores of all items, including those related to clinical impairment or distress. The researcher then diagnoses food addiction based on the symptom scores and clinical significance criteria (mild = 2–3 symptoms plus impairment or distress, moderate = 4–5 symptoms plus impairment or distress, severe = 6 or more symptoms plus impairment or distress). In this study, the C-mYFAS2.0 demonstrated excellent reliability, with a Cronbach's alpha coefficient of 0.873.

### Social physique anxiety scale

*Xu (2003)* developed the Chinese version of the Social Physique Anxiety Scale which contains 3 dimensions: anxiety for social comparison (*e.g.*, item 3, "I feel comfortable even though other people's body size is superior to mine"), discomfort for self-expression (*e.g.*, item 2, "I consistently experience unease and discomfort when I expose my body to others"), and concerns for others' evaluation (*e.g.*, item 6, "I am worried that others will make fun of my body size when I socialize with them"). The authors of the scale have made the scale publicly available and free to use (*Xu, 2003*). All items were scored on a 5—point Likert scale, with scores ranging from 1 to 5 indicating "strongly disagree" to "strongly agree". The total score ranges from 15 to 75, with higher scores indicating a greater tendency toward social physique anxiety. The Cronbach's alpha coefficient for this scale in this study was 0.877.

### Expressive suppression scale

The Expressive Suppression Scale is one of the subscales of the Emotion Regulation Strategies Scale, a scale that is widely used worldwide and has been licensed by the original authors for use in all Chinese populations (*Gross & John, 2003*; *Zhao et al., 2015*). It uses a 7—point Likert scale with scores ranging from 1 to 7 indicating "Strongly Disagree" to "Strongly Agree," respectively. This subscale has four items (*e.g.*, item 9, "I don't express sadness and anger when I feel it") and the total score ranges from 4 to 28, with higher scores indicating that participants tend to suppress their thoughts. The Cronbach's alpha coefficient for this subscale in the current study was 0.834.

### Social avoidance and distress scale

The Social Avoidance and Distress Scale (SADS) consists of 28 items, of which 14 are used to assess social avoidance (*e.g.*, "I typically feel at ease when conversing with the opposite sex") and the remaining items to measure social distress (*e.g.*, "I remain composed and at ease even in unfamiliar social settings") (*Watson & Friend, 1969*). The SADS version we used here was revised by Peng et al. The original authors have licensed the scale for use

 

in all Chinese populations (*Peng, Fan & Li, 2003*). All items in the scale are assessed by "yes" or "no", with a score of 1 if the answer is "yes "and 0 if the answer is "no". The total score ranges from 0 to 28, with higher scores representing more severe social avoidance and distress in individuals. The Cronbach's alpha coefficient for the scale in the study was 0.910.

## Data and statistical analysis

Initially, we exported the data from Questionstar into an Excel format. Two researchers independently screened the data, and in cases of uncertainty, a third researcher reached a consensus. Subsequently, the screened data were imported into SPSS 26.0 for analysis. In the initial phase, we employed Harman's single-factor analysis to assess the presence of common method bias in the study. This involved performing an exploratory factor analysis (EFA) on all variables. A common method bias would be indicated if a single factor emerged or if any factor explained more than 40% of the variance. The EFA revealed nine factors with eigenvalues exceeding 1, and the first factor accounted for 11.84% of the variance, well below the critical threshold of 40%, suggesting no significant common method bias.

In the subsequent phase, we conducted a descriptive analysis of all the variables of interest and evaluated their internal consistency using Cronbach's alpha coefficient, considering values above 0.7 as acceptable. Additionally, Pearson correlation analysis was performed to examine the relationships between the variables.

For the final phase, to enhance the validity of our tests, we standardized all scale data using z-scores. Post-standardization, the data adhered to a standard normal distribution, with half of the observations below zero and the other half above zero, and the variables exhibited a mean of 0, a standard deviation of 1, and variability ranging from −1 to 1.

In the final analytical step, we employed the PROCESS macro (Version 3.4) with Model 6 to examine the mediating effects of expressive suppression and social avoidance and distress between social physique anxiety and food addiction. We designated food addiction as the dependent variable and social physique anxiety as the independent variable. We sequentially tested whether expressive suppression (H1) and social avoidance and distress (H2), as well as their combined effects (H3), could mediate the relationship between the independent and dependent variables. The significance of all paths was determined by 95% bias-corrected bootstrapped confidence intervals (based on 5,000 samples), which did not include 0.

# RESULTS

## Descriptive statistics

As shown in Table 1, of the 1,151 students included in this study, 25.54% were males, 63.08% were living in the city, and the mean age was 21.44 ($SD = 4.77$). 57.52% of the students had a bedtime of 22: 00–24: 00, 58.56% of the participants were undergraduate students, and 93.74% of the students were non-smokers.

Means and standard deviations for psychometric variables are also presented in Table 1. The score of food addiction was 1.34 ($SD = 2.26$). If the hierarchical diagnostic approach
**Table 1 Descriptive statistics of demographic characteristics and food addiction, social physique anxiety, expressive suppression, and social avoidance and distress (N = 1,151).**

| Variable | Mean (SD) or number (%) |
|---|---|
| *Gender* | |
| Male | 294 (25.54%) |
| Female | 857 (74.46%) |
| *Age (Years)* | 21.44 (4.77) |
| *BMI* | |
| <18.5 | 230 (19.98%) |
| 18.5–23.9 | 631 (54.82%) |
| 24–27.9 | 118 (10.25%) |
| >28 | 172 (14.94%) |
| *Home location (lives now)* | |
| City | 726 (63.08%) |
| Country | 426 (36.92%) |
| *Bedtime (Recent year)* | |
| Before 22: 00 | 59 (5.13%) |
| 22: 00–24: 00 | 662 (57.52%) |
| After 24: 00 | 430 (37.36%) |
| *Education* | |
| 3–year college student | 60 (5.21%) |
| Undergraduate college student | 674 (58.56%) |
| Postgraduate college student | 417 (36.23%) |
| *Cigarettes per day (Recent year)* | |
| No smoking | 1079 (93.74%) |
| <10 cigarettes 1 day | 55 (4.78%) |
| 10–20 cigarettes 1 day | 5 (0.43%) |
| >20 cigaettes 1 day | 12 (1.04%) |
| **Food addiction** | 1.34 (2.26) |
| **Social physique anxiety** | 43.21 (10.52) |
| **Expressive suppression** | 14.66 (5.61) |
| **Social avoidance and distress** | 13.68 (7.47) |

Notes.
  Abbreviations: BMI, Body Mass Index.

is followed, A total of 80 (6.90%) subjects were regarded as having food addiction with the number of mild, moderate and severe food addiction are 9 (0.70%), 17 (1.40%) and 54 (4.60%) respectively. The scores of the Social Physical Anxiety Scale, Expressive Suppression Scale, and Social Avoidance and Distress Scale of the participants were 43.21 ($SD = 10.52$), 14.66 ($SD = 5.61$), and 13.68 ($SD = 7.47$), respectively.

## Correlation analyses

Correlation coefficients for all variables are presented in Table 2. Positive correlations were found between food addiction, social physique anxiety, expressive suppression, and social avoidance and distress ($P$ all < 0.001), while gender was positively correlated with social physique anxiety ($r = 0.062$, $P < 0.05$) and social avoidance and distress ($r = 0.073$,

**Table 2  Correlation coefficients for the study variables ($N = 1,151$).**

| Variable | 1 | 2 | 3 | 4 | 5 | 6 | 7 | 8 | 9 |
|---|---|---|---|---|---|---|---|---|---|
| 1. Food addiction | 1 | | | | | | | | |
| 2. Social physique anxiety | 0.288[**] | 1 | | | | | | | |
| 3. Expressive suppression | 0.163[**] | 0.219[**] | 1 | | | | | | |
| 4. Social avoidance and distress | 0.203[**] | 0.490[**] | 0.195[**] | 1 | | | | | |
| 5. Gender[a] | −0.049 | 0.062[*] | −0.142[**] | 0.073[*] | 1 | | | | |
| 6. Cigarettes per day[a] | 0.096[**] | −0.016 | 0.001 | −0.059[*] | −0.240[**] | 1 | | | |
| 7. Bedtime[a] | 0.147[**] | 0.095[**] | −0.009 | 0.098[**] | 0.010 | 0.083[**] | 1 | | |
| 8. Education[a] | 0.011 | −0.101[**] | −0.075[*] | −0.117[**] | 0.007 | 0.076[*] | 0.054 | 1 | |
| 9. BMI[a] | 0.034 | 0.034 | 0.073[*] | −0.061[*] | −0.208[**] | 0.030 | 0.002 | 0.071[*] | 1 |

**Notes.**

Abbreviations: BMI, Body Mass Index.

[**]$P < 0.01$.

[*]$P < 0.05$.

[a]Indicates point-biserial correlation coefficient.

$P < 0.05$) and negatively correlated with expressive suppression ($r = -0.142, P < 0.01$), the cigarettes per day was positively correlated with food addiction ($r = 0.096, P < 0.01$) and negatively correlated with social avoidance and distress ($r = -0.059, P < 0.01$), bedtime was positively correlated with food addiction ($r = 0.147, P < 0.01$), social physique anxiety ($r = 0.095, P < 0.01$), and social avoidance and distress ($r = 0.098, P < 0.01$), education was negatively correlated with social physique anxiety ($r = -0.101, P < 0.01$), expressive suppression ($r = -0.075, P < 0.05$) , and social avoidance and distress ($r = -0.117, P < 0.01$), and BMI was positively correlated with expressive suppression ($r = -0.073, P < 0.05$) and negatively correlated with social avoidance and distress ($r = -0.061, P < 0.01$).

## Path analysis

As shown in Table 3 and Fig. 2, all the coefficients of paths were statistically significant after variables such as gender, cigarettes per day, bedtime, education, and BMI were controlled for. The social physique anxiety positively and significantly influenced expressive suppression ($\beta = 0.224, P < 0.001$). When social avoidance and distress was included, the result showed the social physique anxiety ($\beta = 0.456, P < 0.01$) and the expressive suppression ($\beta = 0.102, P < 0.001$) significantly and positively predicted the social avoidance and distress, and ultimately, the food addiction was able to be predicted significantly and positively by social physique anxiety ($\beta = 0.226$ , $P < 0.001$), expressive suppression ($\beta = 0.097, P < 0.001$) and social avoidance and distress ($\beta = 0.075, P < 0.05$).

The results of the total, direct and indirect effects of the serial mediating effect are shown in Table 4. Four pathways produced the effects. Path1: social physique anxiety → food addiction; Path2: social physique anxiety → expressive suppression → food addiction; Path3: social physique anxiety → social avoidance and distress → food addiction; Path4: social physique anxiety → expressive suppression → social avoidance and distress → food addiction. The results showed that the total indirect effect (0.058) accounted for

**Table 3** Direct and indirect effects of social physical anxiety on food addiction ($N = 1{,}151$).

| | Consequent | | | | | |
| | Expressive suppression | | Social avoidance and distress | | Food addiction | |
| Antecedent | B (SE) | t | B (SE) | t | B (SE) | t |
|---|---|---|---|---|---|---|
| Social physique anxiety | 0.224 (0.028) | 7.778*** | 0.456 (0.026) | 17.289*** | 0.226 (0.032) | 6.977*** |
| Expressive suppression | | | 0.102 (0.026) | 3.859*** | 0.097 (0.029) | 3.332*** |
| Social avoidance and distress | | | | | 0.075 (0.032) | 2.315* |
| Gender | −0.354 (0.069) | −5.515*** | 0.077 (0.062) | 1.231 | −0.077 (0.068) | −1.127 |
| Cigarettes per day | −0.079 (0.084) | −0.930 | −0.122 (0.075) | −1.623 | 0.239 (0.082) | 2.907** |
| Bedtime | −0.025 (0.169) | −0.831 | 0.065 (0.027) | 2.409* | 0.116 (0.029) | 3.925*** |
| Education | −0.085 (0.272) | −1.756 | −0.099 (0.043) | −2.293* | 0.061 (0.047) | 1.299 |
| BMI | 0.040 (0.177) | −1.284 | −0.077 (0.028) | −2.751** | 0.012 (0.031) | 0.397 |
| $R^2$ | 0.078 | | 0.266 | | 0.123 | |
| F | 16.054*** | | 59.079*** | | 19.941*** | |

**Notes.**
Abbreviations: BMI, Body Mass Index.
***$P < 0.001$.
**$P < 0.01$.
*$P < 0.05$.

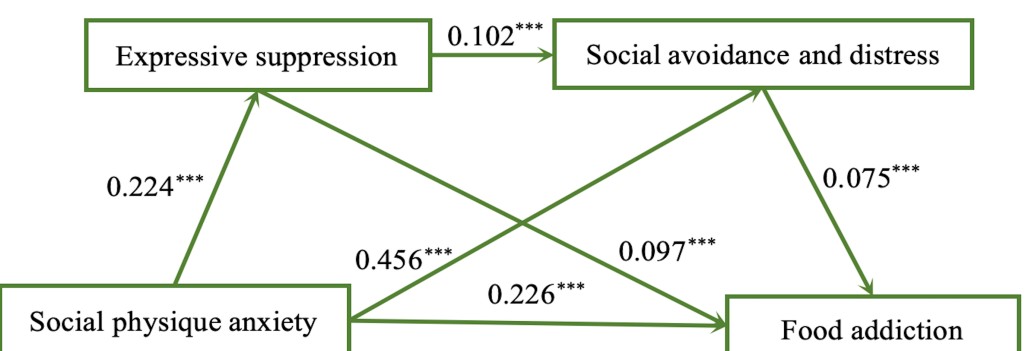

**Figure 2** Direct and indirect of effects of social physique anxiety on food addiction. Note : *** $P < 0.001$, ** $P < 0.01$, * $P < 0.05$.

20.40% of the total effect (0.284) and 25.70% of the direct effect (0.226) in the relationship between social physique anxiety and food addiction. This result indicates that 20.40% of social physique anxiety's positive effect on food addiction works through three effects. Specifically, the effects are (a) the mediating effect of expressive suppression, (b) the mediating effect of social avoidance and distress, and (c) the mediating effect of expressive suppression and social avoidance and distress, respectively. The mediating effect (a), (b) and (c) represented 7.70%, 12.30% and 0.70% of the total effect, respectively, and 9.70%, 15.50% and 0.90% of the direct effect, respectively. The mediating effect of social avoidance and distress was significantly stronger than expressive suppression or serial mediation in the relationship between social physique anxiety and food addiction.

**Table 4  Mediating effects of expressive suppression and social avoidance and distress between social physique anxiety and food addiction ($N = 1,151$).**

| | Effect | SE | Boot LLCI | Boot ULCI | Ratio of indirect to total effect | Ratio of indirect to direct effect |
|---|---|---|---|---|---|---|
| Total effect | 0.284 | 0.028 | 0.228 | 0.339 | – | – |
| Direct effect | 0.226 | 0.032 | 0.162 | 0.290 | – | – |
| Total indirect effect | 0.058 | 0.015 | 0.030 | 0.087 | 0.204 | 0.257 |
| Social physique anxiety → Expressive suppression → Food addiction | 0.022 | 0.008 | 0.008 | 0.038 | 0.077 | 0.097 |
| Social physique anxiety → Social avoidance and distress → Food addiction | 0.035 | 0.013 | 0.009 | 0.060 | 0.123 | 0.155 |
| Social physique anxiety → Expressive suppression → Social avoidance and distress → Food addiction | 0.002 | 0.001 | 0.001 | 0.004 | 0.007 | 0.009 |

## DISCUSSION

This research is, to the best of our knowledge, the first one exploring the potential impact of expressive suppression and social avoidance and distress on the connection between social physique anxiety and food addiction in a comprehensive survey of Chinese university students. The results validated all three propositions concerning the possible indirect routes involving expressive suppression and social avoidance and distress, comprising the independent influence of expressive suppression (H1), the independent influence of social avoidance and distress (H2), and a sequence of effects of expressive suppression and social avoidance and distress (H3). The hypothesis test effect values for H1 and H3 were small (both $R^2 < 0.04$), while the effect value for H2 was 0.035, approaching the threshold for a medium effect size ($R^2 = 0.04$) (*Ferguson, 2009*).

The current study elucidates that, within the cohort of Chinese college students, social physique anxiety harbors a positive correlation with food addiction. The empirical evidence gleaned from structural equation model illustrates that social physique anxiety can exert a direct and positive influence on food addiction. This relationship may be attributed to the fact that students burdened with social physique anxiety frequently grapple with an amalgamation of unsettling emotions including distress, trepidation, and despondency in their quotidian existence, leading to volatile mood states (*Herring et al., 2021*). The heightened anxiety spawned by social physique anxiety often precipitates erratic conduct, typified by overzealous dietary restrictions. Concurrently, in the face of negative emotional experiences, these individuals are susceptible to losing self-regulatory control and succumbing to disordered eating practices. This outcome further corroborates previous research that delineates the persistent correlation between social physique anxiety and eating disorders (*Alcaraz-Ibáñez, Paterna & Griffiths, 2023*). It also implies that interventions targeting body size-related cognitions can effectively mitigate food addiction. For instance, reshaping individuals' perceptions of others' evaluations regarding body size and clarifying the irrelevance of others' perceptions can be particularly beneficial.

The current investigation reveals that expressive suppression serves as a partial intermediary in the dynamic between social physique anxiety and food addiction

amongst Chinese university students, aligning with antecedent research wherein expressive suppression acted as a conduit between adverse emotional states and food addiction (*Mitchell & Wolf, 2016*). Those afflicted by social physique anxiety grapple with a diminished self-perception and allure, rooted in dissatisfaction and apprehension regarding their body image, thereby resorting to emotional suppression with greater frequency (*Davison & McCabe, 2006*). The reliance on expressive suppression to manage negative emotions frequently proves ineffective (*Kraft et al., 2023*). Consequently, individuals may resort to pathological eating behaviors, such as binge eating, as a means to alleviate the inadequacy of their emotional regulation efforts in handling negative feelings (*Lavender & Anderson, 2010*; *Muehlenkamp et al., 2012*). In other words, social physique anxiety propels college students towards the habitual employment of emotional suppression, a strategy ill-suited for adaptive emotional management, leading to recurrent indulgence in eating as a means to enhance mood. Overall, these findings underscore the mediating role of expression suppression between negative body image perceptions and unhealthy eating behaviors. It indicates that targeting expression suppression can serve as an effective intervention to mitigate the impact of social physique anxiety on food addiction. Nevertheless, due to the limitations of our research instrument for the emotion regulation strategy, which was a subscale, we were unable to determine whether adaptive emotion regulation strategies, such as cognitive reappraisal, produce effects that are contrary to or resemble those of expression suppression.

This investigation elucidated that social avoidance and distress serves as a conduit in the nexus between social physique anxiety and food addiction, that is to say, social physique anxiety precipitates food addiction through the intermediary of social avoidance and distress. The genesis of social physique anxiety within an individual engenders a trepidation regarding adverse judgments from their peers, thereby undermining their capacity to authentically present themselves (*Zartaloudi et al., 2023*). This, in turn, detrimentally impacts their communicative competencies, subsequently impairing their social interaction capabilities. Concurrently, extant literature has uncovered those tendencies towards social seclusion and diminished social support networks can exacerbate food addiction severity (*Lacroix & von Ranson, 2021*; *Li et al., 2022*). On one hand, social avoidance and distress prompt individuals to decrease opportunities for group dining, which is crucial in reducing binge eating behavior and promoting adherence to healthy eating norms (*Cao, Zhu & Meng, 2021*). On the other hand, social avoidance and distress results in deficient personal social support, isolating individuals and depriving them of the comfort and shared coping strategies typically available when eating disorders emerge or are imminent (*Ma et al., 2021*). Ergo, the diminishment of social engagement resultant from social avoidance and distress precipitates a decline in social support, fostering the progression towards food addiction. Prevention and early intervention for social avoidance and distress can significantly mitigate the entrenchment of eating disorders in emerging adults (*Ramjan et al., 2024*). Therefore, it is crucial to diversify the forms of social support interventions and enhance the comprehensiveness of social support sources. Person-centered therapies are recommended to foster a deeper self-understanding and a holistic approach to life,

encompassing social, cultural, ecological, and spiritual connections (*Garcia, Cloninger & Cloninger, 2023*).

Moreover, expressive suppression and social avoidance and distress serve as intermediary conduits linking social physique anxiety and food addiction. Notably, the framework advanced by *Morrison & Heimberg (2013)* posits that individuals exhibiting excessive concern regarding others' unfavorable assessments are prone to construe ambiguous social cues negatively. To circumvent undesirable negative evaluation or social ostracism, individuals apprehensive of negative evaluations tend to inhibit the articulation of their thoughts while resorting to non-adaptive strategies such as expressive suppression (*Goodman et al., 2021*). This maladaptive approach predisposes a pervasive state of despondency, thereby exacerbating negative affectivity and antisocial behaviors. This observation lends credence to Clark and Wells' social anxiety framework, which posits that individuals with self-critical perceptions are inclined to construe social contexts as fraught with peril, precipitating anxious manifestations and tendencies towards social withdrawal (*Rapee & Heimberg, 1997*). Thus, employing expressive suppression as an emotion regulation strategy intensifies social avoidance and distress. An individual's adverse social conduct and negative social affect can precipitate a decline in personal social reinforcement and a transformation in emotional requisites, prompting recurrent engagement in detrimental practices such as excessive food consumption as a compensatory measure for their emotional deficiencies (*Yang et al., 2023*). This finding elucidates a potential mechanism through which social physique anxiety contributes to food addiction and underscores the importance of intervening with effective emotion regulation strategies for individuals experiencing this anxiety. Encouraging positive emotional expression can mitigate social withdrawal. Simultaneously, providing individuals with abundant social resources can expand their social networks and decrease the incidence of eating disorders.

## IMPLICATIONS AND LIMITATIONS

The present investigation yields both theoretical and pragmatic implications. Firstly, it broadens the extant body of knowledge concerning the determinants of food addiction, specifically elucidating the impact of socio-physical appearance on individual dietary preferences, moderated by the overlay of non-adaptive emotion regulation strategies and social disorder. Thus, forging a synthesis of emotion regulation theory, the social anxiety framework, and the negative affect reinforcement hypothesis. Pragmatically, these insights furnish cogent approaches to mitigating food addiction, thereby forestalling related physique or psychological pathologies. In particular, pedagogical initiatives in educational institutions could anchor on body shape perception, empowering students to adopt a balanced perspective on the influence of physique on personal development, assuaging negative affect associated with corporeal dissatisfaction, and bolstering self-assurance in body image through active engagement in physical activities and structured guidance on body sculpting. For students experiencing social physique anxiety, the propensity towards social appearance anxiety and consequent social reticence may be attenuated through the cultivation of efficacious and salubrious emotion regulation strategies, thereby diminishing their food addiction.

Nevertheless, this investigation is encumbered by several limitations. Initially, this study encompassed Chinese college students and employed a self-exclusion method for survey administration, potentially constraining the generalizability of the results. Moreover, the self-exclusion approach may have been excessively permissive. Future research ought to broaden the sample population and utilize precise measurement instruments to standardize the inclusion criteria more rigorously, thus augmenting the credibility of the study. Secondly, the employment of a cross-sectional design precludes the establishment of causality between variables of interest. Prospective studies could adopt longitudinal methodologies, such as cross-lagged analysis, to elucidate these relationships more definitively. Thirdly, a disproportionate representation of female participants, who constituted 74.46% of the study population and previous studies demonstrated a higher incidence of food addiction compared to their male counterparts (*Carr et al., 2017*), could bias the results. Males experiencing different emotional adversities exhibited increased food addiction levels relative to females (*Hoover et al., 2022*; *Hoover et al., 2023*). It is noteworthy that the present study did not address the impact of personality on food addiction, despite the fact that individuals with eating disorders often exhibit specific personality traits. For instance, anorexics typically display a high degree of self-direction to maintain restrictive eating habits, whereas bulimics often exhibit significant impulsivity (*Garcia et al., 2022*). Neglecting to consider heritable antecedents of anxiety may have introduced bias into the findings, suggesting that the observed relationship between anxiety and food addiction may not be as robust as reported. Future research could incorporate temperament and personality questionnaires to assess the personalities of individuals with eating disorders, thereby mitigating the potential interference of personality traits on the correlation between other factors and food addiction. Finally, self-reported data may be subject to social desirability and self-presentation biases. Future studies should use social desirability control scales to detect and adjust for these biases. Additionally, employing multiple data collection methods, such as behavioral observations or physiological measures, can help validate self-reported outcomes.

## CONCLUSION

The present study confirmed the mediating effects of expressive suppression, social avoidance and distress between social physique anxiety and food addiction. Based on these findings, we suggest that emerging adults should be supported in developing body image confidence to reduce social appraisal fears. Additionally, individuals should be encouraged to engage in positive emotional expression and enhance their social support to lower social barriers, ultimately alleviating food addiction. However, this study has some limitations. Future research should consider the role of personality and well-being factors in the relationship between social physique anxiety and eating disorders. It is also recommended to increase the sample size and use longitudinal studies to improve the reliability and generalizability of the findings.

### Ethical statement

This study was approved by the Ethics Committee of Qingdao Mental Health Center (QDJWZXWZLL2024002), and all procedures of the human participants involved in this study followed the ethical standards of the institution and/or the National Research Committee, as well as the Declaration of Ethical Standards of the Helsinki Research Committee of 1964 and its subsequent amendments or similar ethical standards. In addition, the work has not been previously published or is not being considered for publication elsewhere. This article has been reviewed and approved by all authors. All authors declare that they have no conflicts of interest.

### Funding

The authors received no funding for this work.

### Competing Interests

The authors declare there are no competing interests.

### Author Contributions

- Yan Li conceived and designed the experiments, performed the experiments, analyzed the data, prepared figures and/or tables, and approved the final draft.
- Yuxia Fan conceived and designed the experiments, performed the experiments, analyzed the data, prepared figures and/or tables, and approved the final draft.
- Jing Lin conceived and designed the experiments, performed the experiments, analyzed the data, authored or reviewed drafts of the article, and approved the final draft.
- Shaobo Shi conceived and designed the experiments, performed the experiments, analyzed the data, authored or reviewed drafts of the article, and approved the final draft.

### Human Ethics

The following information was supplied relating to ethical approvals (i.e., approving body and any reference numbers):

The Ethics Committee of Qingdao Mental Health Center approved the study (QDJWZXWZLL2024002).

### Data Availability

The raw measurements are available in the Supplemental Files.

### Supplemental Information

Supplemental information for this article can be found online at http://dx.doi.org/10.7717/peerj.17910#supplemental-information.

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
