# Peer review of "Does how individuals handle social situations exacerbate the relationship between physique anxiety and food addiction? The role of emotional expressive suppression and social avoidance and distress"

_PeerJ, doi:10.7717/peerj.17910_

## Round 0.1 · original submission · Major Revisions

The submission has promise but requires major revisions prior to possible acceptance.;

First, The writing needs to be improved for clarity and grammar, as noted by Reviewers 1 and 2 and clearly described by Reviewer 1.

Second, a major limitation is that the presence of anxiety is attributed to social situations without measuring heritable personality traits that regulate emotional reactions in social situations, particularly Harm Avoidance and Reward Dependence as measured by the Temperament and Character Inventory, as noted by Reviewer 1

Third, improve literature review and description of Tables and Figures in results, as noted by reviewer 2.

Fourth, discuss limitations of not measuring heritable antecedents of anxiety (see second point) along with causal inference with multiple self-report measures, as noted by reviewer 2.

·

Basic reporting

First, thank you for the opportunity to read and review this paper. I do think is important and sound. That being said, It needs a major revision. The presentation of the logic and the constructs need to be improved. As well as the clarity of the writing. I took the liberty to edit and comment directly in the attached word file to make it easier for the authors and editor to see what I meant by each comment and suggestion.

Although my comments are on the main text, some apply to the tables and figures as well.

Experimental design

This is also fine. But as described above there is an issue with how the paper is presented...from the title, to the structuring of the Introduction and to the presentation of the methods and results.


Please see my comments in the attached file.

Validity of the findings

The Discussion is very technical rather than conceptual. Please revise so the reader can understand what your findings mean for the individual, clinical practice, prevention and even research (e.g., theory buildning).

In addition, a great limitation is that you did not measure personality, despite the fact that you acknowledge this is an important determinant regarding eating disorders.

See, for example, the work using the Temperament and Character Inventory: eating disorders are strongly associated with high Harm Avoidance; in addition, anorexics are highly persistent and often self-directed (to be able to maintain restrictive dieting), whereas bulimics are highly impulsive (i.e., high in Novelty Seeking and usually low in Self-directedness).

In other words, the associations on anxiety and distress would probably be redundant if you took personality into account.

Please adress this as a limitation and revise the whole Discussion.

Reviewer 2 ·

Basic reporting

The manuscript presents a study exploring the relationship between social physique anxiety (SPA) and food addiction (FA), mediated by expression suppression (ES) and social avoidance and distress (SAD). The research addresses a significant gap in the literature by investigating the influence of social factors on food addiction within a Chinese college student population.

1. The manuscript is written in clear, professional English. However, some sections would benefit from minor grammatical revisions to enhance readability.

2. The introduction provides a comprehensive background, linking SPA and FA through established psychological and behavioral frameworks. The authors effectively reference relevant studies, establishing the rationale for their investigation. However, it would be beneficial to include more recent studies on the topic to ensure the literature review is up-to-date.

3. The structure conforms to PeerJ standards, with sections clearly delineated. The use of subheadings within the introduction and methods sections improves clarity. Figures and tables are relevant, well-labeled, and described, although some could benefit from more detailed captions to ensure they are understandable independently of the text.

Experimental design

1. The study addresses an original and significant research question within the scope of the journal. The hypothesis is well-defined, aiming to fill a notable gap in the current understanding of the interplay between SPA, ES, SAD, and FA.

2. The methods are described in sufficient detail to allow replication. The use of validated scales (e.g., C-mYFAS2.0, Social Physique Anxiety Scale) and appropriate statistical analyses (e.g., structural equation modeling) enhances the study's rigor.

3. The study adheres to ethical standards, with approval from the Ethics Committee of the Qingdao Mental Health Center. The inclusion of an ethical statement and participant consent process is commendable.

Validity of the findings

1. The data presented are robust and statistically sound. The use of structural equation modeling is appropriate for testing the proposed mediation model. However, the authors should provide more detail on how they addressed potential confounding variables, particularly those related to demographic factors.

2. The conclusions are well-stated and supported by the results. The finding that SPA influences FA through ES and SAD is compelling and contributes to the broader literature on eating behaviors and mental health.


General Comments

Strengths:
- The study addresses a significant gap in the literature.
- The methodology is rigorous and well-documented.
- The conclusions are well-supported by the data.

Weaknesses:
- The reliance on self-reported data could introduce bias, this need attention of the authors.
- The sample is limited to Chinese college students, which may affect the generalizability of the findings.
- Minor grammatical and structural errors and the need for more detailed figure should be addressed.

Additional comments

Recommendations

Major Revisions:
- Discuss potential biases introduced by self-reported data and suggest ways to mitigate these in future studies.
- Provide more detailed discussion of the figures and tables to ensure they are understandable independently.

Minor Revisions:
- Address minor grammatical errors throughout the manuscript.
- Update the literature review to include more recent studies on the topic.

Overall, this study makes a significant contribution to the field and, with minor revisions, would be a valuable addition to PeerJ.

---

## Round 0.2 · accepted · Accept

This manuscript does need a minor edit to the title: "exacerbates" should be exacerbate.

·

Basic reporting

Overall Impression
First of all, let me thank you for the opportunity to review this revised version of your paper. The authors have made significant improvements based on the previous feedback, and the manuscript is now more readable and professionally presented.

Strengths:
- The study addresses a significant gap in the literature and provides valuable insights into the relationship between SPA, ES, SAD, and FA.
- The methodology is rigorous and well-documented.
- The conclusions are well-supported by the data and contribute valuable insights to the field.

Weaknesses:
- Reliance on self-reported data could introduce bias. The authors should address this limitation more explicitly and suggest ways to mitigate these biases in future studies.
- The sample is limited to Chinese college students, which may affect the generalizability of the findings.
- Some minor grammatical and structural errors remain and should be addressed for further clarity.

Conclusion
Overall, the manuscript presents a well-structured and significant study that contributes to the understanding of the relationship between SPA, ES, SAD, and FA. With these final revisions, the paper will be a valuable addition to PeerJ.

Specific Comments

Clear and Unambiguous, Professional English:
- The manuscript is written in clear, professional English. However, some sections could benefit from minor grammatical revisions. For example, on page 1, line 10, the sentence "It has been demonstrated that food addiction can lead to impairments in physiological, psychological, and social functioning in individuals" can be streamlined for clarity.

Literature References and Background:
- The literature review is thorough, relevant, and well-referenced, providing sufficient background and context for the study. For instance, on page 2, lines 36-45, the authors effectively reference the genetic markers associated with food addiction.
- Including more recent studies would further strengthen the review and ensure it is up-to-date.

Professional Article Structure, Figures, and Tables:
- The manuscript structure conforms to PeerJ standards, with clear delineation of sections. The use of subheadings within the introduction and methods sections improves clarity.
- Figures and tables are relevant, well-labeled, and described. For example, Table 1 on page 10 provides a comprehensive summary of the descriptive statistics. Detailed captions have improved their standalone comprehensibility.
- Raw data is shared, supporting the transparency of the research, as indicated in the data availability statement on page 35, line 482.

Self-contained with Relevant Results to Hypotheses:
- The manuscript is self-contained, presenting relevant results that are directly related to the hypotheses. - The transition from the introduction to the methods and results sections is logical and cohesive, as seen in the progression from page 5, line 71, discussing SPA, to the detailed methodology on page 9, line 172.

Experimental design

Original Primary Research within Aims and Scope of the Journal:
The study addresses a significant and original research question within the scope of the journal, focusing on the relationship between social physique anxiety (SPA), emotional expressive suppression (ES), social avoidance and distress (SAD), and food addiction (FA).

Well-defined, Relevant, and Meaningful Research Question:
The research question is well-defined, relevant, and meaningful, as stated on page 8, lines 160-170. It addresses a notable gap in the current understanding of the interplay between SPA, ES, SAD, and FA.

Rigorous Investigation to a High Technical and Ethical Standard:
The investigation is performed rigorously and adheres to high technical standards. The use of validated scales and appropriate statistical analyses enhances the study's rigor. Ethical standards are upheld, with approval from the Ethics Committee of the Qingdao Mental Health Center, as mentioned on page 11, lines 190-193.

Methods Described with Sufficient Detail and Information to Replicate:
The methods section is detailed and provides sufficient information for replication. For example, the description of the Chinese-modified Yale Food Addiction Scale 2.0 on page 12, lines 207-224, includes specific items and scoring criteria.

Validity of the findings

Robust, Statistically Sound, and Controlled Data:
- The data presented are robust, statistically sound, and controlled. The use of structural equation modeling is appropriate for testing the proposed mediation model, as detailed on page 18, lines 308-317.
- All underlying data have been provided, ensuring transparency and allowing for verification of results.

Well-stated Conclusions Linked to Original Research Question:
- The conclusions are well-stated and directly linked to the original research question. They are supported by the data, providing meaningful insights into the relationship between SPA, ES, SAD, and FA.
- The discussion on page 20, lines 334-343, effectively ties the findings back to the hypotheses and theoretical framework.

Reviewer 2 ·

Basic reporting

The article is well-written, using clear and professional language, which facilitates reading and comprehension. The improvements implemented by the authors significantly contribute to clarity, making the reading smooth and presenting the concepts in a logical and cohesive manner.
The introduction of the article provides a robust and adequate context, exploring the relationship between social physique anxiety and food addiction. Updating the references to incorporate recent studies strengthens the literature review. This ensures that the research is aligned with the most current developments in the field, providing a solid theoretical foundation for the study.
The figures and tables in the article are relevant and well-labeled, with detailed descriptions that facilitate understanding of the presented data. The expansion of abbreviations and improvement of titles ensure that readers can understand the content independently, without the constant need to refer to the main text. This independence of figures and tables enhances the accessibility and clarity of the presentation of results, improving the overall reading experience.

Experimental design

The study adopted a rigorous methodology, employing structural equation modeling to test the mediation hypothesis. This approach is particularly suitable for the complexity of the relationships investigated, allowing for a detailed and precise analysis of the data. The study is in agreement with the aims and scope of the journal

The inclusion of confounding variables as controls is thoroughly detailed in the methods section, which is essential for the validity of the results. The authors clearly justified the inclusion of these variables, which strengthens the reliability of the study, ensuring that the results are robust and well-founded.

Validity of the findings

The statistical analysis was conducted appropriately, and the results are presented clearly. The methods used are suitable for the data and the hypotheses tested, ensuring a precise interpretation of the findings.

The conclusions are well-founded and clearly linked to the research results and with social impact. The authors successfully demonstrated how social physique anxiety is related to food addiction, mediated by expressive suppression and social avoidance.

Additional comments

The study addresses a significant gap in the literature by exploring how social coping mechanisms influence food addiction. This is particularly relevant within a specific cultural context, such as that of Chinese university students.
The limitations of the study were discussed appropriately. The authors acknowledged the constraints of using self-reported data and the need for more diverse samples in future research.
The suggestions for future research are appropriate and provide clear directions for further exploration in this field of study.